# Impact of Heat Stress on Meat Quality and Antioxidant Markers in Iberian Pigs

**DOI:** 10.3390/antiox10121911

**Published:** 2021-11-29

**Authors:** Zaira Pardo, Ignacio Fernández-Fígares, Manuel Lachica, Luis Lara, Rosa Nieto, Isabel Seiquer

**Affiliations:** Departamento de Fisiología y Bioquímica de la Nutrición Animal, Estación Experimental del Zaidín, Consejo Superior de Investigaciones Científicas (CSIC), San Miguel 101, 18100 Armilla, Granada, Spain; zaira.pardo@eez.csic.es (Z.P.); ignacio.fernandez-figares@eez.csic.es (I.F.-F.); manuel.lachica@eez.csic.es (M.L.); luis.lara@eez.csic.es (L.L.); rosa.nieto@eez.csic.es (R.N.)

**Keywords:** meat, Iberian pig, antioxidant enzymes, meat quality

## Abstract

Heat stress is associated with impaired meat quality and disruption of redox balance. This study investigated the effect of chronic exposure to high temperature on meat quality and antioxidant markers of muscles (*longissimus lumborum* and *gluteus medius*) of growing Iberian pigs. Twenty-four pure Iberian pigs were allocated during 28 days to one of three treatments (*n* = 8/treatment): thermoneutral conditions (20 °C) and ad libitum feeding (TN), heat stress conditions (30 °C) and ad libitum feeding (HS) and thermoneutral and pair-fed with HS (TN-pf). Muscles of the HS group had greater intramuscular fat content than the TN-pf group and higher Zn levels than TN and TN-pf, whereas differences on fatty acid composition were negligible. Heat exposure did not affect pH, color coordinates of redness (a*) and yellowness (b*) and MDA values but had a positive influence on lightness and drip losses. Moreover, chronic heat stress stimulated the activity of antioxidant defenses SOD, CAT and GPx. The statistical factor analysis adequately classified the muscles studied, but was unable to differentiate samples according with treatments. Findings of the present study support an adaptive response of the Iberian pig to high temperatures and show the high Iberian meat quality even under adverse climate situations.

## 1. Introduction

Global warming is one of the main threats facing the world in the 21st century, causing frequent heat waves and increasing the global temperature of the planet [1]. High ambient temperature leads to heat stress that affects animal health prompting a severe challenge for livestock production, especially in the warmer parts of the world [2]. The heat stress not only impacts on physiological changes and growth performance in live animals, but can also compromise meat quality characteristics such as pH, water holding capacity or meat color [3], resulting in economic losses for producers and lower acceptance by consumers [4,5]. This fact has especial importance considering that livestock products will have to increase 40% by 2050 as a result of the 33% expected increase of world population, according to the United Nations Food and Agriculture Organization [6].

Pigs are quite sensitive to high temperatures because they have scattered sweat glands and very limited capacity to dissipate heat [7]. To decrease metabolic heat production, pigs tend to decrease their feed intake, affecting growth and performance parameters [8]. In addition, persistent heat stress has a great impact on muscle metabolism and may decrease the meat quality of pigs [5,9]. This impact has been associated with increased oxidative reactions and production of reactive oxygen species (ROS) thus disrupting the redox balance that ensures the stability in skeletal muscle and preserve the meat quality [10,11]. Compositional changes of meat such as decreased intramuscular fat (IMF) have also been reported as a result of high temperature exposition [12,13]. Traditionally, these changes have been related to decreases of feed intake, but certain studies have shown that heat stress per se may also reduce metabolic rate and alter oxidative metabolism in muscle to reduce thermogenesis [13]. Currently, it remains unclear what changes affecting quality of porcine meat are dependent or not on decreased nutrient supply.

The Iberian pig (*Sus mediterraneus*) is an autochthonous breed from the Iberian Peninsula that generates products of outstandingly high quality very appreciated in specialized markets [14]. Although Iberian pigs are rustic animals, their abundant subcutaneous fat could make them more vulnerable to heat stress than lean breeds, as thicker subcutaneous fat layer delays heat dissipation [15]. In addition, the main pig producing areas in Spain face hot-summer Mediterranean climate [16] according to Köppen classification, characterized by high temperatures during summer with maximum averages of 32–36°C. Despite the economic importance and organoleptic traits of Iberian products, there is a lack of comprehensive studies on the effect of high temperature exposure on the quality of Iberian pig meat. 

With this background, and given the current need of identifying pigs less susceptible to heat stress that would raise the efficiency of pig industry in the face of climate changes [17], the objective of the present work was to evaluate the effects of prolonged high temperature exposure on meat quality traits and antioxidant capacity of Iberian pigs. To address this purpose, two muscles, *longissimus lumborum* (glycolytic) and *gluteus medius* (glycolytic and oxidative) were examined (hereinafter referred as *longissimus* and *gluteus*), as representatives of the commercial pieces more valuables and appreciated by consumers.

## 2. Materials and Methods

### 2.1. Animals and Experimental Design

All experimental procedures and animal care were in agreement with Spanish Ministry of Agriculture guidelines (RD53/2013) based on European legislation for the care and use of animals in research (EU Directive 2010/63/EU for animal experiments). The experiment authorization was approved by the Bioethical Committee of the Spanish National Research Council (CSIC, Spain) and the competent authority (Junta de Andalucía, Spain, project reference 28/06/2016/118).

A total of 24 pure Iberian barrows supplied by Sanchez Romero Carvajal Jabugo S.A (Puerto de Santa María, Cádiz, Spain) were involved in the study. At arrival pigs were individually housed in 2-m^2^ slatted pens (with individual feeders and nipple drinkers) in thermoneutral conditions. Water was provided ad libitum during the entire trial. After one-week acclimation period, pigs were weighted (44.0 ± 1.36 kg) and assigned to one of the three treatments (*n* = 8 per group): (1) thermo-neutral (20 °C) and fed ad libitum (TN); (2) heat stress (30 °C) and fed ad libitum (HS); (3) thermo-neutral (20 °C) and pair-fed (TN-pf) to HS, to differentiate the direct effects of heat exposure from those due to decrease in feed intake. The feed intake for TN-pf group was calculated daily based on averaged feed intake of the HS group the previous day. Two separated temperature-controlled rooms (20 °C and 30 °C) were used; each room’s temperature was recorded every 15 min with the aid of a data logger (HOBO UX100-011; Onset Computer Corporation, Bourne, MA, USA). Photoperiod was established to 12 h of artificial light (8:00 to 20:00 h) and 12 h of darkness.

All pigs were fed the same diet based on barley, corn and soy bean meal (146 g crude protein/kg, 8.9 g lysine/kg and 16.6 MJ metabolizable energy/kg), supplemented with essential amino acids to maintain an adequate amino acid profile and cover all nutrients requirements [18]. Feed refusals were collected, weighed and dried in order to calculate daily feed intake.

After four weeks of experimental period pigs were slaughtered following an over-night fast, by electrical stunning and immediate exsanguination. A scheme of the experimental design is depicted in Figure 1.

### 2.2. Muscle Quality Traits

Evaluation of meat quality was conducted as described in a previous study [14] following the reference methods [19].

The right half carcass was used for muscles determinations. Instantly after slaughter, aliquots of *longissimus* and *gluteus*, clean of connective tissue and superficial fat, were cut into small pieces, dipped in liquid nitrogen and stored at −80 °C until measurements of muscle antioxidant status. At 30 min postmortem (p.m.), a portable pH meter (HI 99163, Hanna instruments, Romania) equipped with a penetration electrode was used to determine the pH values of *longissimus* (at the last rib level) and *gluteus* (pH 30 min). The carcasses were kept in a cold room at 4 °C for 24 h, after which the pH was measured again (pH 24 h). Then, the complete *longissimus* and the *gluteus* were separated from the carcass. A 3-cm thick steak from the *longissimus* and the *gluteus* were allowed to blooming (15 min, 4 °C) and used for color measurement. Afterwards, these samples were vacuum-packed and stored at −20 °C for chemical composition analysis. Furthermore, 2 cm steaks were cut from *longissimus* muscle, trimmed of external fat and connective tissue and used to determine drip loss, thawing loss, and cooking loss. Samples for thawing loss were weighed and immediately frozen (−20 °C). 

#### 2.2.1. Physical Meat Quality Assessment

Meat color was measured using a Minolta Chroma Meter (CR-400, Konica Minolta Corp., Japan) with illuminant D65 and 0° standard observed, in agreement with the CIE L*, a*, b* color system. The apparatus was previously calibrated with white ceramic tile and the color coordinates were measured as the average of 3 random readings: L* (lightness, from 0, dark, to 100, white), a* (redness) and b* (yellowness). Moreover, Chroma (C*) and the hue angle (h°), described as color intensity and saturation or tone, respectively, were calculated by using the following equations: C = (a*^2^ + b*^2^)^0.5^ and h° = arctg b*/a*.

For determining water holding capacity of *longissimus*, meat slices were weighed and placed within a closed plastic container on a supporting mesh; after 24 and 48 h at 4 °C samples were re-weighed to calculate drip loss. The frozen slices were thawed for 24 h at 4 °C, slightly blotted dry and weighed to calculate thawing loss. For measuring cooking loss, a meat slice previously weighed was placed in a plastic bag and cooked in a hot water bath until reaching 72 °C using a temperature probe with a penetration perforator (LCD Digital Thermo Hygrometer, DC105). Thereafter, cooked samples were placed on crushed ice to cool to a temperature of 10 °C, blotted dry and weighed. Water losses were calculated as a percentage of the initial weight.

#### 2.2.2. Chemical Composition

Muscles aliquots for nutrient and mineral analysis were previously grounded (Retsch GM 200, Germany), lyophilized (freeze dryer Virtis Genesis, SQ25EL) and homogenized with liquid nitrogen (Retsh ZM 200).

Dry matter (method 934.01) and total ash content (method 942.05) were determined using official methods [20]. IMF was extracted with chloroform: methanol 2:1 and quantified by Soxhlet [20]. Total nitrogen was analyzed by the Dumas procedure using a LECO Truspec CN equipment (LECO Corporation, St. Joseph, MI, USA) and protein content was calculated using the factor of 6.25. Gross energy was determined in an isoperibolic bomb calorimeter (Parr Instrument Co., Moline, IL, USA).

The mineral content of muscles (Fe and Zn) was analyzed according with the procedure described by Palma-Granados et al. [21]. Briefly, representative aliquots of muscles were completely digested with concentrated HNO_3_:HClO_4_ (1:4) and by heating to high temperatures (180–220 °C) in a sand beaker (Block Digestor Selecta S-509; J.P. Selecta, Barcelona, Spain). Fe and Zn were analyzed by flame atomic absorption spectroscopy (FAAS) in a Perkin-Elmer Analyst 700 Spectrophotometer (Norwalk, CT, USA), using certified external standards (European Commission, Reference Materials Unit, Geel, Belgium) to assess the accuracy of the method: bovine liver (BCR 185R) for Zn and lyophilized brown bread (BCR 191) for Fe. All glassware and polyethylene sample bottles used for mineral analysis were washed with 10 mM nitric acid and demineralized water (Milli-Q Ultrapure Water System, Millipore Corp., Bedford, MA, USA) was used throughout the study. 

#### 2.2.3. Fatty Acid (FA) Analysis

Firstly, fat was extracted by the method of Folch et al. [22] and FA were then methylated according to Kramer and Zhou [23], using HCl/methanol for obtaining the fatty acid methyl esters (FAME). Pentadecanoic acid (C15:0; Sigma-Aldrich, Madrid, Spain) was used as internal standard. FAME were identified with a gas chromatograph equipped with a flame ionization detector (Focus GC, Thermo Scientific, Milan, Italia) and using a 100 m × 0.25 mm × 0.2 µm capillary silica gel column (TR-CN100 Teknokroma, Barcelona, Spain). The temperature of the program was 70 to 240 °C and the injector and detector were maintained at 250 °C. The carrier gas used was helium at a flow rate of 1 mL/min. Individual FAME peaks were identified by comparing their retention times with those of standards (47885-U, Sigma Aldrich) and results were expressed as a percentage of the total FAME identified.

### 2.3. Antioxidant Status 

#### 2.3.1. Lipid Peroxidation

The determination of the oxidative stability of the muscle samples was carried out using thiobarbituric acid-reactive substances (TBARS) assay, according to Seiquer et al. [14]. Briefly, muscles samples (0.5 g) were homogenized with 5 mL of 0.15 M KCl + 0.1 mM BHT (30 s, 4 °C) and centrifuged. Aliquots of supernatant were incubated with 1% (*w*/*v*) 2-thiobarbituric acid in 50 mM NaOH and 0.25 mL of 2.8% (*w*/*v*) trichloroacetic acid for 10 min at 100 °C. The chromogen was extracted with *n*-butanol and absorbance was measured spectrophotometrically at 535 nm (Pharmaspec UV 1800, Shimadzu, Kyoto, Japan). Concentration of TBARS was determined using a standard curved prepared with 1,1,3,3-tetramethoxypropane and expressed as mg malondialdehyde (MDA)/kg muscle.

#### 2.3.2. Antioxidant Activity

To study the antioxidant capacity of the muscle samples the ABTS (2,2-azinobis-(3-ethylbensothiazoline)-6-sulfonic acid) and DPPH (2,2-diphenyl-1-picrylhydrazyl) assays (for measuring the free radical scavenger activity) and the FRAP method (for assessing the ferric reducing antioxidant power) were performed. A chemical extraction was carried out previous to analysis of antioxidant assays in muscle samples. Samples (250 mg) were mixed with 2.5 mL of acidic methanol/water (50:50 *v*/*v*, pH 2), shaken at 220 rpm for 60 min (circulating shaker OVAN, Barcelona, Spain) and centrifuged at 2500 rpm for 10 min at 4 °C (Sorvall RC 6 Plus centrifuge, Thermo Scien-tific, Madrid, Spain). Supernatant was recovered and a second extraction was performed with acetone/water (70:30, *v*/*v*, 2.5 mL) to obtain the final chemical extract. The procedures were conducted as described previously [24], using 96-well microplates and by reading the absorbance in a Victor X3 multilabel plate reader (Waltham, MA, USA). The results were expressed as mM of Trolox equivalents per kg of muscle, using aqueous solutions of Trolox 0.01–0.1 mg/mL for the calibration curve.

In the ABTS assay, the ABTS• + solution was obtained by mixing 2.45 mM potassium persulfate with ABTS 7 mM 12–16 h before use. This solution was diluted with 5 mM phosphate buffered saline to an absorbance of 0.70 ± 0.02 at 750 nm. Twenty µL of muscle extract were added to 280 µL of ABTS solution and incubated 20 min in the dark before reading the absorbance at a 750 nm. 

For the DPPH method 50 µL of the muscle extract were mixed with 250 µL of DPPH solution (74 mg/L in methanol prepared daily). After 60 min of incubation period, the absorbance was read at 520 nm maintaining the temperature in the measurement chamber at 30 °C. 

The FRAP reagent was prepared daily by mixing 10 mM ^−^2,4,6-Tri(2-pyridyl)-1,3,5-triazine (TPTZ) with 40 mM HCl, 20 mM ferric chloride and 0.3 M acetate sodium buffer (pH 3.6) in a ratio 1:1:10 *v*/*v*/*v*. Twenty µL of muscle extract were added to 280 µL of warmed FRAP reagent (37 °C), incubated at 37 °C in darkness for 30 min and the absorbance was read at 595 nm. 

#### 2.3.3. Antioxidant Enzyme Activity 

The activities of catalase (CAT), superoxide dismutase (SOD) and glutathione peroxidase (GPx) were assessed in aliquots of *longissimus* and *gluteus* muscles. The samples (5 g muscle) were homogenized in 2 mL of ice-cold of sucrose buffer (0.32 M, pH 7) using an Ultra-Turrax^®^ homogenizer (IKA-Werke GmbH & Co. KG, Staufen, Germany). Homogenates were centrifuged at 4 °C for 10 min at 20,000× *g* and the supernatant fractions were collected and used to determine the activity of the antioxidant enzymes according with Pardo and Seiquer [25] using a UV spectrophotometer (Pharmaspec UV 1800, Shimadzu, Kyoto, Japan). 

CAT activity was measured by monitoring the H_2_O_2_ decomposition as a consequence of the action of the enzyme, by spectrophotometric measurement at 240 nm. For the analysis of SOD activity, the technique is based on the generation of superoxide radicals using the xanthine/xanthine oxidase system. In the presence of superoxide radical, a reduction of cytochrome c occurs, which is spectrophotometrically monitored at 550 nm. GPx activity was determined by the instantaneous formation of glutathione oxidized during the reaction catalyzed by GPx; this reaction is coupled with the reuse of reduced glutathione using glutathione reductase and NADPH. The oxidation of NADPH is indicative of GPX activity and is monitored at 340 nm.

### 2.4. Statistical Analysis

Analyses were performed in triplicate. The data obtained were analyzed by applying analysis of variance (two-way ANOVA) to study the effects of the treatment (TN, TN-pf and HS) and the type of muscle (*longissimus* and *gluteus*) as the main factors, and their interaction. LSD test was used to compare mean values and significant differences were established at *p* < 0.05). Data are presented as means and standard error of the mean (SEM).

The relationships between the different variables were evaluated by Pearson’s coefficient. In addition, with the aim of evaluate the contribution of the different variables in the samples classification, a factor analysis procedure was applied. Preliminary analyses with all the variables (7 corresponding to chemical composition, 7 of quality traits, 7 of oxidative status and 22 variables of FA profile, with a total of 43 variables) were carried out to select those with the highest weight in the classification. Varimax rotation was applied to the 24 variables with the greatest weight as an attempt to clarify the relationship among factors and explore their impact in the samples differentiation. Furthermore, using the new factors as dimensions, the graph representation allows assessing the similarity of the samples according with treatments or types of muscle. 

All statistical calculations were carried out using the StatGraphics Centurion XVI software version 16.1.18 (StatPoint Technologies Inc., Warrenton, VA, USA). 

## 3. Results and Discussion

The temperature recorded during the 28 d of experimental period was on average 19.9 ± 0.20 °C and 30.2 ± 0.20 °C for thermo-neutral and heat stress conditions, respectively. The feed intake was significantly reduced (by 20%) among Iberian growing pigs submitted for 3 weeks to elevated temperature (average values of 2931, 2342 and 2248 g /d in the TN, HS and TN-pf groups, respectively, expressed on dry matter basis), supporting findings of previous bibliography in lean breeds [3].

### 3.1. Chemical Composition and FA Profile of Muscles 

The chemical composition of muscles is depicted in Table 1. 

The muscle composition has a strong impact on the nutritional and organoleptic properties of porcine meat. Particularly, the IMF content and FA profile have essential effects on the oxidative stability, tenderness, juiciness and flavor [26]. In addition, the higher IMF level and the content of oleic acid (C18:1n 9) are considered as differential quality traits of Iberian pig products compared with conventional breeds [27].

In the present study the different composition of the two muscles studied was clearly manifested, being the gluteus richer in the nutritional components, including minerals (especially Fe), than the longissimus. Differences in IMF content and composition traits between longissimus and gluteus medius muscles in pigs have been previously documented [14,28]. Such changes have been attributed to the genes involved in the differentiation of muscle cells as well as in carbohydrate and lipid metabolism, which are overexpressed in the gluteus muscle and may be initially caused by differences in the body location, function, and metabolism of the two porcine muscles [28].

The prolonged exposure to high temperatures in the present study caused significant differences in the meat composition regarding dry matter, IMF, energy and Zn content, whereas no effect on protein and ash levels were observed. It should be remarked that differences between TN-pf and HS groups are indicative of a direct effect of the ambient temperature, without the confounding effect of the different intake, whereas differences with the TN group would also suggest an influence of the feed consumption. At thermo-neutral conditions, the reduced intake had a negative effect on the IMF level of muscles of Iberian pigs, while the heat stress had a positive effect at the same intake level, thus counteracting the decrease due to the lower feed consumption and restoring the IMF level to values observed in TN ad libitum fed pigs. This finding is no consistent with previous studies in conventional porcine breeds that have shown that both feed restriction and exposure to high temperature reduced IMF deposition [12,13]. In lean pigs the effect of high temperature on IMF content was mainly explained by the decreased feed intake [29,30]. However, Ma et al. [13], using a similar experimental design than in the present assay, showed that high temperature has an effect per se in the regulation of genes related to muscle structure and involved in the adipocytokine signaling, thus affecting directly meat quality, apart from indirect effects from depressed feed intake. Moreover, Xin et al. [31] show that, at similar level of feed intake, heat stress reduces the amount of acetyl coenzyme A and fatty acid synthase in the longissimus muscle of pigs and also inhibit beta-oxidation of FA by decreasing the hydroxyacyl CoA dehydrogenase. 

The differential effect found in Iberian pigs in the present assays regarding IMF in muscle could suggest a compensatory mechanism in the activity of enzymes and the expression of genes involved, trying to restore the lower IMF consequence of the reduced intake. Lu et al. [32] have shown that IMF differences in muscle of broilers under chronic heat stress are linked to increased mRNA expressions of fatty-acid synthase (FAS) and acetyl-CoA carboxylase, enzymes participating in the FA synthesis. In fact, the activity and gene expression of lipogenic enzymes, such as FAS, malic enzyme and glucose-6-phosphate dehydrogenase, closely related with IMF deposition, are higher in Iberian than in lean pigs [14], and could be specifically stimulated in response to heat stress.

The fat content of muscles is highly associated to the moisture and energy levels, which was supported by the strong correlations between these components (*p* < 0.001) found in the present assay (Appendix A). The effect of high temperatures in the mineral content of meat has been scarcely studied, with the only exception of Fe, due to its relationships with color pigments. In the present study, the heat stress had no significant effect on the Fe content of muscles, but interestingly, provoked an increase of the Zn level not described previously, which was independent of the restricted feed intake. The reasons of the Zn higher content in meat of Iberian pigs under heat stress are unknown, but may be a positive nutritional factor for the consumer, since Zn intake is essential for human health due to its role in enzymatic systems, cell division and growth, gene expression, immune and reproductive functions and antioxidant defenses [33].

The FA composition of muscle lipids also has important repercussion in the meat quality, since it is linked to sensorial and technological aspects that affect the consumer acceptability [27] and because polyunsaturated (PUFA) to saturated (SFA) FA ratio in one of the most reliable markers of nutritional value of meat [34]. Higher amounts of PUFA and increased ratios of PUFA/SFA were found in gluteus compared with longissimus (Table 2), according with previous data indicating that muscles with a higher proportion of oxidative fibers have a greater ability to accumulate specially *n*-6 FA [35]. However, effects of high temperature on FA profile of porcine meat have received little attention. In lean pigs, it has been reported increased levels of monounsaturated FA (MUFA) in longissimus and lower ratio PUFA/SFA in gluteus after exposure to hot ambient [12]. In the present study, ambient temperature had a very mild effect in the FA composition of Iberian muscles and only slight decreases of the n3 FA and the PUFA/SFA values in the gluteus of HS group compared with TN-pf were observed. The stability of the FA composition of Iberian muscles under adverse temperature conditions could be an important factor regarding susceptibility to lipid oxidation, which is strongly influenced by saturation degree of IMF and the high proportion of PUFA [36]. Moreover, oleic acid content is a differential characteristic of Iberian pig products, and the fact that oleic acid was not influenced by heat stress represents an added benefit in terms of quality for the consumer market [37].

### 3.2. pH, Color and Water Holding Capacity

The pH and color coordinates of muscles of growing Iberian pigs exposed to the different treatments are depicted in Table 3. Figure 2 shows the water losses of the *longissimus* muscle.

Values of pH were very similar in *longissimus* and *gluteus*, both at 30 min and 24 h p.m. However, strong differences of color were observed between muscles, being *gluteus* darker and redder that *longissimus*, according with the metabolic pattern of the oxidative (Type I, red muscle) and glycolytic (Type II, white muscle) fibers, that is closely related to myoglobin concentration [38]. Color differences found in the present assay agree with those described for *gluteus* and *longissimus* muscles in previous studies [35].

In the current experiment, the hot exposure did not have significant effects on pH values (initial or ultimate) and color coordinates of redness (a*) and yellowness (b*) and the related indexes of C* and h°. Nevertheless, muscles of the HS Iberian pigs had reduced lightness (L*), i.e., were darker, than those of the TN group. No significant differences were observed among HS and TN-pf, suggesting that the decreased feed intake might have contributed to the difference in meat lightness. 

Exposure to high temperatures in pigs has been usually associated with a deterioration of meat quality and occurrence of PSE (pale, soft, exudative) meat, with decreased pH values, pale color and increased drip losses [4,11,39,40], which has been explained by a stimulation of muscle glycogenolysis and further metabolic acidosis under heat conditions. However, increased values of ultimate pH have also been observed in other heat stress assays [12,30]. The decline of pH p.m. subsequent to the glycogen catabolism and the lactate and hydrogen ions accumulation is critical in determining the quality of meat and is strongly related to color and water holding capacity. Any deviation of the normal pH rate could lead to poor meat quality, producing PSE or DFD (dark, firm and dry) meat, in cases of low or high pH values at 24 h p.m., respectively [5]. Conversely to our results, higher lightness (paler color) and lower a* and b* values have been found as effects of high temperature exposure in muscles of lean pigs [11,30,40], which has been attributed to the muscle oxidative status and the gradual transformation of the different states of the pigment myoglobin, the principle responsible for meat color, that is very sensitive to oxidation [41]. Discoloration of muscles results from oxidation of both ferrous myoglobin derivatives to ferric iron of metmyoglobin [41] and therefore, no occurrence of pale color could indicate a higher protection against oxidation in HS muscles (as discussed below). In addition to the redox state, the pigment content and the Fe level (located in the heme ring of the myoglobin) may account for much of the variation of a* values found at high temperature ambient [42]. In the present assay, the similar color in muscles among treatments was probably linked to the lack of differences in Fe content, which was supported by the significant relationships between Fe and color coordinates, especial with redness (*p* < 0.001, Appendix A). Since consumer purchasing decisions greatly depend on the product color, it results interesting from a market point of view that meat color of Iberian pigs at heat stress conditions maintain the stability and redness, which is one of its hallmarks.

Typically, heat stress has been associated in pigs with a more exudative meat and lower water holding capacity, which has an adverse effect on nutritional properties and tenderness of meat [5,11]. Studies performed in autochthonous breeds also have shown that high environmental temperature negatively affect drip loss and shear force of meat [39]. Interestingly, in the current study muscles of Iberian pigs exposed to high temperatures had lower drip losses (i.e., higher water holding capacity) at 24 and 48 h than those of the TN group, with no significant effects on cooking and thawing losses (Figure 2). The TN-pf group showed intermediate values, indicating a partial influence of the reduced feed intake. Lower drip loss in pork have been associated to higher abundance of heat shock proteins (HSPs), a group of proteins that pigs synthesize to cope up with the adverse changes such as the heat shock [43] and that play a crucial role in water retention and postmortem meat quality [44]. Thus, the positive reaction observed in the present study concerning quality traits of muscles of Iberian pigs, could be related with a greater HSPs production in response to heat stress.

### 3.3. Antioxidant Markers

Markers of oxidative status are shown in Table 4. 

According with previous studies, heat stress accelerates the oxidation of muscle tissue causing changes in the pro-oxidant/antioxidant balance and compromising meat quality in conventional pigs [11] and broilers [45,46].

The MDA is a biomarker of lipid peroxidation, one of the most important causes of meat deterioration, and MDA level has been used to determine the extent to which heat stress cause free radical damage in skeletal muscle [10]. Elevated MDA values has been reported in heat-stressed lean pigs as a consequence of a pro-oxidant cellular environment and excessive free radical production [3,11]. However, exposure to hot ambient of Iberian pigs in the present assay did not cause apparent changes in the level of muscle MDA, since no global effect of the treatment was seen (*p* > 0.05). The lack of effect in MDA values may be related with the absence of differences in the FA profile, since differences in lipid peroxidation could be consequence of higher content of PUFA, more likely to be oxidized [35].

Antioxidant properties, determined by ABTS, DPPH and FRAP, measure the radical scavenging activity and the reducing power in tissues and have been used previously to screen the antioxidant capacity and the redox balance of different porcine muscles [47]. In the present assay, the heat exposition led to increased levels of DPPH compared with TN and TN-pf groups in *longissimus* muscle and counteracted the drop due to lower intake (indicated by significant differences between TN and TN-pf groups) observed in the reducing-power (FRAP) of the *gluteus* muscle. No differences due to treatment were observed in the ABTS assay. 

In addition to the level of antioxidant activity, the free radical injury described as a consequence of heat stress may be due to a failure of the antioxidant defense system to adequately respond to stress conditions; therefore, we measured the activity of the main antioxidant enzymes, SOD, CAT and GPx. The SOD acts as a first line of defense catalyzing the conversion of superoxide to hydrogen peroxide and functions in conjunction with CAT and GPX to convert it into water and molecular oxygen [48]. Unexpectedly, a stimulation of the enzymatic defense system was found in muscles of heat-stressed pigs, with higher activity of the antioxidant enzymes compared with TN-pf (CAT significantly increased in both muscles and SOD tended to be higher in *longissimus*) and also compared with TN (GPx significantly augmented in *gluteus*). On the contrary, lower activity [11] or no effect [30] of the antioxidant enzymes have been found in muscles of conventional pigs exposed to prolonged heat stress (21–30 days). 

It has been shown that the production of SOD and CAT compromises during heat stress, which results in increasing MDA and oxidative stress [11,32,45,49]. The apparent absence of oxidative damage (stable MDA levels) found in muscles of Iberian pigs in the current assay could stem from a positive response in antioxidant enzymes activity, serving to mitigate the heat stress induced-free radical injury. Thus, the probably initial insult seems to be compensated by the antioxidant defense activity, resulting in the resolution of oxidative damage. Interestingly, proteomic studies in lean pigs have shown that there is an initial defensive response of muscle against heat stress by inducing antioxidant-proteins expression, although unable of avoid deterioration of meat quality [40]. Likewise, Rosado-Montilla et al. [10] have detected a transient mechanism in pigs subjected to acute heat stress (35 °C) during 1 or 3 days, since CAT and SOD activity in muscle were significantly increased after 1 day and returned to thermoneutral levels by Day 3. The most effective response in Iberian pigs was supported by the enhanced radical-scavenging ability and reducing power in muscles of animals under heat stress, as indicated by the higher DPPH and FRAP values. Previous studies have shown that feeding diets containing high levels of antioxidant compounds led to increased values of ABTS and DPPH in muscles of broilers [45] and Iberian pigs [50], which is accompanied by an enhanced muscle antioxidant capacity [45]. However, the effect of heat stress on the antioxidant status of porcine muscles has not been studied before.

Finally, significant differences were detected in the antioxidant markers between the muscles studied, with the only exception of SOD activity, although in some cases (MDA, ABTS and DPPH) interactions between treatment and type of muscle were detected. Such variances confirm the different oxidative stability of *longissimus* and *gluteus* due to the different nature of their fibers and based in significant differences in the mRNA expression patterns, which evidence that the transcriptomic profile of the skeletal muscle tissue is affected by anatomical, metabolic and functional factors [28].

### 3.4. Factor Analysis

To reduce the variables studied into a small number of factors and explore its contribution to the differentiation of the muscles, a factor analysis using a varimax rotation was applied. The analysis performed showed eight factors with eigenvalues > 1, that accounted for 81.7% of total variance (F1 27.2%, F2 14.8%, and F3-F8 from 11.9 to 4.2%). The F1 was mainly defined by the instrumental color (L*, −0.728; a*, 0.926; b*0.835; C*, 0.939), Fe content (0.719) and antioxidant markers (MDA, 0.521; CAT −0.698; GPx, 0.527). Moreover, the main variables defining Factor 1 were statistically correlated (*p* < 0.05, Appendix A). The nutrient composition clearly contributed to F2 (IMF, 0.929; energy, 0.934; dry matter, 0.893; ash, −0.766) and F3 was specially affected by FA composition (MUFA, 0.947; C18:1n9, 0.941; PUFA, −0.685; SFA, −0.533). Therefore, variables defining F1 and F2 had a major contribution to the differentiation of each group of samples. The graphic representation of Factors 1 and 2 illustrates potential relationships between the samples analyzed according to the main variables affecting each factor (Figure 3), by projecting the data points into a bi-dimensional space. Firstly, we explored the classification of the muscles (Part A) and a clear differentiation between them was observed, particularly taking into account F1, represented horizontally, and to a lesser extent considering F2, represented vertically: samples of *longissimus* were located in the left lower side and those of *gluteus* in the right upper side, being both muscles completely distinguished practically without overlap or crossover. Secondly, we represented the samples identifying the treatments (part B), TN, TN-pf and HS, and no clear distinction among the different groups was shown, indicating a high level of similarity between the samples of the three treatments. We repeated the data representation considering F1 and F3 and similar results were obtained (graphic not shown). Therefore, the factor analysis showed that the muscle samples could not be correctly classified according to the treatments, which means that the chronic exposure to high temperatures did not lead to significant differentiation of the muscles of Iberian pigs according with the variables analyzed of nutritional composition, quality traits and antioxidant markers.

## 4. Conclusions

The present study shows for the first time the impact of chronic heat stress (28 d of exposure at 30 °C) on the quality and antioxidant markers in two muscles of Iberian pigs. It was observed that the prolonged heat exposure did no compromise meat quality traits of Iberian pigs, and even certain improvements, such as in IMF content, lightness and drip losses were shown. In addition, a positive response on some markers of oxidative status in heat-stressed pigs was detected, as MDA level was not altered, and antioxidant capacity was stimulated. Therefore, the stimulation of the antioxidant defenses may be suggested as a probable mechanism in the resolution of the oxidative damage. The statistical factor analysis showed that the muscles *longissimus* and *gluteus* were correctly classified, but samples from the three treatments were undifferentiated according to the variables analyzed in the present assay. Therefore, it appears that the Iberian pig has a significant capability to cope with high ambient temperatures, in contrast with lean pigs and other autochthonous breeds that suffer a qualitative decrease of meat quality. The molecular mechanisms involved in the adaptability of Iberian pigs to heat stress deserve further studies.

Findings of the present study support a different response to heat stress of the Iberian pig than conventional pigs and confirm the high Iberian meat quality even under adverse situations of climate change.

## Figures and Tables

**Figure 1 antioxidants-10-01911-f001:**
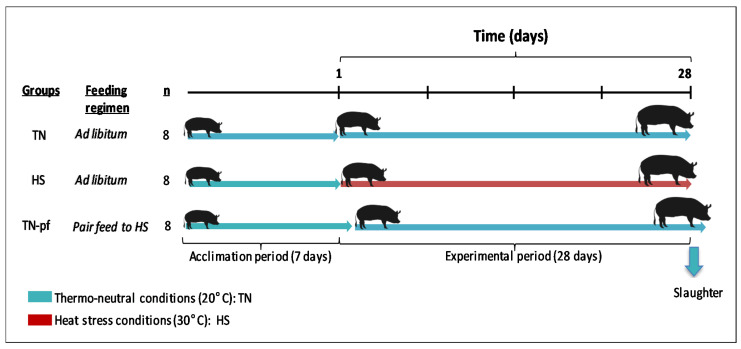
Scheme of the experimental design.

**Figure 2 antioxidants-10-01911-f002:**
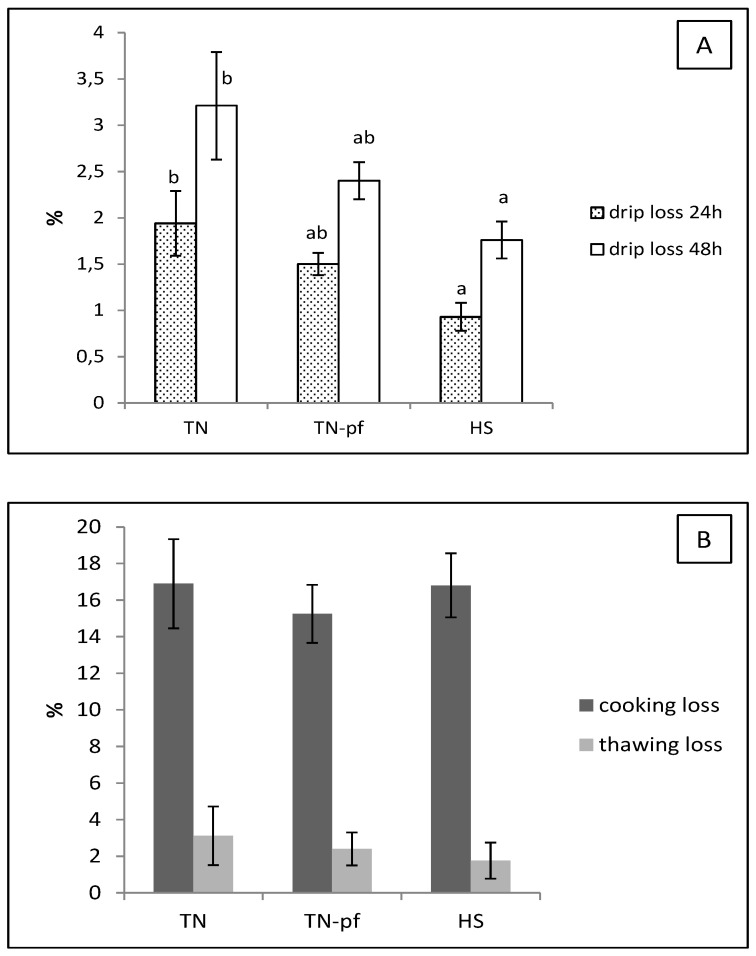
Water losses in *longissimus lumborum* of pure Iberian pigs exposed during 28 days to different ambient temperature. (**A**) Drip loss; (**B**) cooking and thawing loss. TN, pigs reared at thermoneutral conditions (20 °C) and fed ad libitum, TN-pf, pigs reared at thermoneutral conditions (20 °C) and pair-fed to HS group, HS, pig reared at heat stress conditions (30 °C) and fed ad libitum. Different letters indicate significant differences between treatments (*p* < 0.05, one-way ANOVA and LSD test).

**Figure 3 antioxidants-10-01911-f003:**
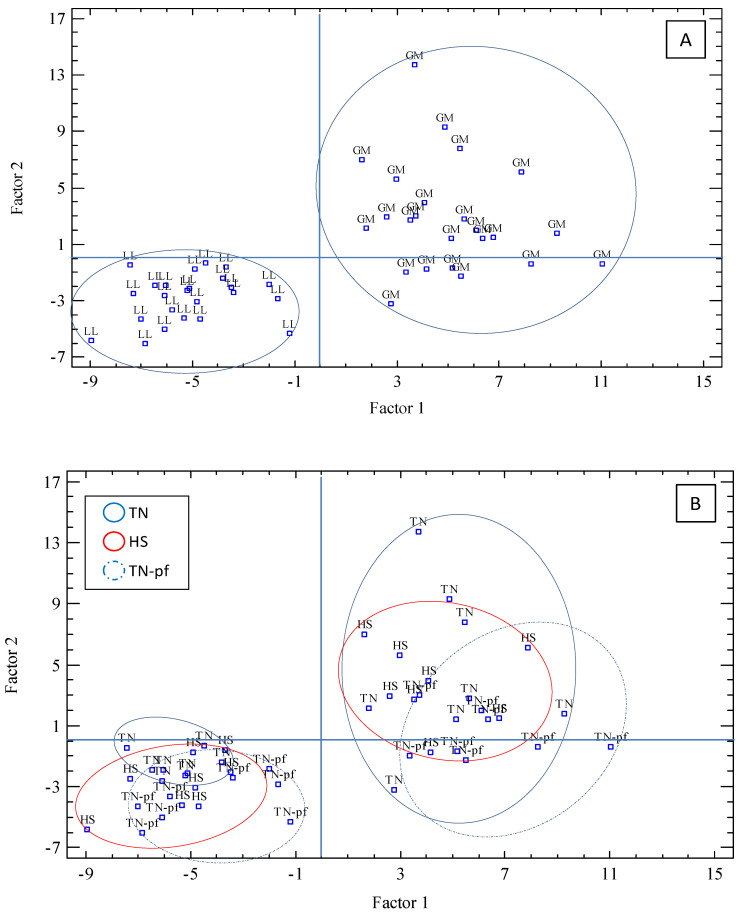
Factor analysis score graph of the two main factors (Factor 1 vs. Factor 2), considering the selected variables analyzed. (**A**) Representation of the different muscles; (**B**) representation of the different treatments. LL, *longissimus lumborum*; GM, *gluteus medius*; TN, pigs reared at thermoneutral conditions (20 °C) and fed ad libitum, TN-pf, pigs reared at thermoneutral conditions (20 °C) and pair-fed to HS group, HS, pig reared at heat stress conditions (30 °C) and fed ad libitum.

**Table 1 antioxidants-10-01911-t001:** Chemical composition in muscles of pure Iberian pigs exposed during 28 days to different ambient temperature.

	*Longissimus*			*Gluteus*				*p*-Value		
	TN	TN-pf	HS	TN	TN-pf	HS	SEM ^1^	Treatment (T)	Muscle (M)	T × M
Dry matter (%)	27.6b	25.9a	26.4a	30.4b	27.9a	30.1b	0.21	0.0008	0.0001	0.2602
Protein (%)	21.6	22.1	21.2	23.6	23.5	23.9	0.35	0.9581	0.0052	0.7369
IMF ^2^ (%)	6.61b	4.77a	5.87b	9.69b	5.91a	8.15b	0.21	0.0027	0.0012	0.4516
Ash (%)	1.97	1.95	2.04	1.59	1.76	1.67	0.05	0.7930	0.0045	0.7226
Energy (kcal/100 g)	170b	153a	160a	196b	168a	189ab	2.3	0.0013	0.0001	0.3963
Fe (mg/kg)	7.05	7.82	7.43	9.08	10.0	10.1	0.28	0.4120	0.0002	0.8893
Zn (mg/kg)	15.4a	14.4a	16.9b	16.2a	15.2a	18.9b	0.26	0.0001	0.0288	0.5486

TN, pigs reared at thermoneutral conditions (20 °C) and fed ad libitum, TN-pf, pigs reared at thermoneutral conditions (20 °C) and pair-fed to HS group, HS, pig reared at heat stress conditions (30 °C) and fed ad libitum. ^1^ SEM: mean standard error, ^2^ IMF: intramuscular fat. Different letters indicate significant differences between treatments in each muscle (*p* < 0.05, two-way ANOVA and LSD test).

**Table 2 antioxidants-10-01911-t002:** Fatty acid profile in muscles of pure Iberian pigs exposed during 28 days to different ambient temperature.

	*Longissimus*			*Gluteus*				*p*-Value		
	TN	TN-pf	HS	TN	TN-pf	HS	SEM ^1^	Treatment (T)	Muscle (M)	T × M
C12:0	0.065	0.054	0.061	0.061	0.053	0.061	0.002	0.1184	0.6624	0.9016
C14:0	1.47	1.39	1.49	1.34	1.31	1.43	0.02	0.1090	0.0313	0.7780
C16:0	27.2	27.4	27.9	26.6	26.7	27.6	0.01	0.2042	0.2051	0.9585
C16:1	3.91	3.73	3.73	3.37	3.36	3.40	0.07	0.8480	0.0074	0.8163
C17:0	0.26	0.26	0.27	0.32	0.31	0.31	0.01	0.9688	0.0223	0.9025
C18:0	12.2	12.2	12.9	12.7	12.1	13.2	0.13	0.0248	0.3586	0.6256
C18:1n9	43.1	41.9	41.0	41.3	41.3	41.6	0.30	0.1717	0.6239	0.5541
C18:1n7	4.31	4.28	4.11	3.98	4.12	3.37	0.10	0.1288	0.0442	0.4759
C18:2n6	4.96	5.99	5.85	6.45	7.40	6.25	0.19	0.1068	0.0057	0.4285
C20:0	0.15	0.15	0.15	0.16	0.14	0.15	0.002	0.0967	0.9927	0.5371
C20:1n9	0.68	0.64	0.66	0.72	0.67	0.75	0.014	0.3346	0.1095	0.6467
C18:3n3	0.20	0.23	0.23	0.25	0.26	0.24	0.011	0.7391	0.1252	0.7529
C20:2n6	0.18	0.19	0.20	0.21	0.24	0.23	0.005	0.0957	0.0002	0.6850
C20:3n6	0.13	0.16	0.15	0.15	0.19	0.14	0.007	0.0977	0.3758	0.5273
C20:4n6	0.79	1.00	0.91	1.07	1.39	0.81	0.06	0.1017	0.1540	0.2925
C20:5n3	0.039	0.056	0.043	0.053	0.085	0.074	0.005	0.1667	0.0245	0.7908
C22:4n6	0.14	0.18	0.16	0.17	0.21	0.15	0.009	0.1267	0.2808	0.5435
C22:5n3	0.10	0.12	0.11	0.13	0.17	0.10	0.006	0.0443	0.1048	0.2836
C22:6n3	0.015	0.018	0.013	0.017ab	0.025b	0.012a	0.0001	0.0116	0.2723	0.3117
SFA ^2^	41.4	41.4	42.8	41.1	40.6	42.8	0.32	0.0659	0.5751	0.8756
MUFA ^3^	52.0	50.6	49.5	50.3	49.4	49.2	0.38	0.1526	0.1677	0.7667
PUFA ^4^	6.54	7.95	7.67	8.51	9.97	8.02	0.25	0.0617	0.0064	0.3169
n6	6.19	7.52	7.27	8.06	9.43	7.59	0.24	0.0655	0.0069	0.3129
n3	0.35	0.43	0.39	0.45a	0.54b	0.43a	0.014	0.0410	0.0048	0.496
PUFA/SFA	0.16	0.19	0.18	0.21a	0.25b	0.19a	0.006	0.0494	0.0064	0.3141
MUFA/SFA	1.27	1.22	1.16	1.23	1.22	1.15	0.02	0.1198	0.6920	0.9242

TN, pigs reared at thermoneutral conditions (20 °C) and fed ad libitum, TN-pf, pigs reared at thermoneutral conditions (20 °C) and pair-fed to HS group, HS, pig reared at heat stress conditions (30 °C) and fed ad libitum. ^1^ SEM: mean standard error, ^2^ SFA: saturated fatty acids, ^3^ MUFA: monounsaturated fatty acids, ^4^ PUFA: polyunsaturated fatty acids. Different letters indicate significant differences between treatments in each muscle (*p* < 0.05, two-way ANOVA and LSD test).

**Table 3 antioxidants-10-01911-t003:** Quality traits in muscles of pure Iberian pigs exposed during 28 days to different ambient temperature.

	*Longissimus*			*Gluteus*				*p*-Value		
	TN	TN-pf	HS	TN	TN-pf	HS	SEM ^1^	Treatment (T)	Muscle (M)	T × M
pH _30 min_	6.26	6.16	6.20	6.30	6.35	6.33	0.04	0.9780	0.167	0.7997
pH _24 h_	5.57	5.56	5.65	5.55	5.51	5.64	0.03	0.2789	0.6580	0.9498
Lightness L*	37.9b	36.6a	36.5a	34.1b	31.7a	32.5a	0.27	0.0190	0.0001	0.6892
Redness a*	7.07	8.29	7.11	12.4	12.8	11.9	0.21	0.1352	0.0001	0.6820
Yellowness b*	3.59	3.70	3.60	6.47	6.20	6.30	0.13	0.9593	0.0001	0.8336
Chroma C*	7.97	9.17	7.82	14.1	14.4	13.5	0.23	0.1337	0.0001	0.7544
Hue angle h°	25.9	25.4	28.3	27.8	27.3	27.8	0.52	0.4130	0.2870	0.5510

TN, pigs reared at thermoneutral conditions (20 °C) and fed ad libitum, TN-pf, pigs reared at thermoneutral conditions (20 °C) and pair-fed to HS group, HS, pig reared at heat stress conditions (30 °C) and fed ad libitum. ^1^ SEM: mean standard error. Different letters indicate significant differences between treatments in each muscle (*p* < 0.05, two-way ANOVA and LSD test).

**Table 4 antioxidants-10-01911-t004:** Oxidative status markers in muscles of pure Iberian pigs exposed during 28 days to different ambient temperature.

	*Longissimus*			*Gluteus*				*p*-Value		
	TN	TN-pf	HS	TN	TN-pf	HS	SEM ^1^	Treatment (T)	Muscle (M)	T × M
MDA ^2^ (mg/kg)	0.17	0.15	0.20	0.23	0.29	0.21	0.04	0.6722	0.0002	0.0205
ABTS (mM/kg)	2.96	2.51	2.64	3.07	3.77	3.45	0.07	0.7480	0.0001	0.0061
DPPH (mM/kg)	0.60a	0.59a	0.65b	0.64	0.65	0.64	0.005	0.0452	0.0061	0.0408
FRAP (mM/kg)	0.42	0.38	0.40	0.51b	0.41a	0.45ab	0.011	0.0304	0.0054	0.4064
GPx ^3^ (U/g prot)	0.24	0.20	0.22	0.14a	0.13a	0.21b	0.008	0.0382	0.0007	0.0752
CAT ^4^ (U/g prot)	332ab	238a	438b	180b	115a	214b	16	0.0016	0.0001	0.4098
SOD ^5^ (U/mg prot)	1.42b	1.27a	1.46b	1.42	1.34	1.41	0.02	0.0668	0.9140	0.6450

TN, pigs reared at thermoneutral conditions (20 °C) and fed ad libitum, TN-pf, pigs reared at thermoneutral conditions (20 °C) and pair-fed to HS group, HS, pig reared at heat stress conditions (30 °C) and fed ad libitum. ^1^ SEM: mean standard error, ^2^ MDA: malondialdehyde, ^3^ GPX: glutathione peroxidase, ^4^ CAT: catalase, ^5^ SOD: superoxide dismutase. Different letters indicate significant differences between treatments in each muscle (*p* < 0.05, two-way ANOVA and LSD test).

## Data Availability

Data is contained within the article or Appendix A.

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
