# Peer review of "Impact of Heat Stress on Meat Quality and Antioxidant Markers in Iberian Pigs"

_antioxidants, 2021, doi:10.3390/antiox10121911_

Round 1

Reviewer 1 Report

This paper, I reviewed, aimed to determine the impact of heat stress on meat quality and antioxidant markers in pigs from Spain.

The topic investigated is of significant interest for pig meat production and animal health.

A very good amount of analytical evaluations have been done. The description of all used methodology results accurate and precise.

The findings have been properly reported and the obtained data well discussed by using the available literature and the recent findings by other authors.

It is of particular interest for readers the evaluation of the mucosa-associated microbiota in broiler chickens fed different dietary treatments.

The Conclusion section provides a clear overview of the findings and their usefulness, evel if it could be further improved.

As specific comments, in order to further improve the quality of the paper, I suggest to:

- An overall check of the English language is suggested in order to further improve the paper's quality.

-Some additional recently published references may add value to the Introduction section;

- Check all acronyms used, spell at first use;

- Conclusion section could be improved.

- The references have been reported in an appropriate form and edited according to the journal’s guidelines.

So, based on my opinion, I think that this paper merits the acceptanceafter minor revision.

Author Response

The authors are grateful with reviewer 1 for the time spending in reviewing the manuscript and the positive comments. All the suggestions made have been taken into account.

The bibliography has been deep revised, and we have added in the Introduction two recent references: Lui et al., 2021, doi.org/10.3390/antiox10101558 and Usala et al., 2021, doi: 10.3389/fgene.2020.612815. The numeration of the references has been changes accordingly.

According with the comments of the referee, the English, all the acronyms and the conclusion section have been checked.

Reviewer 2 Report

In this study, the authors examined the effects of heat stress on oxidative stress and antioxidant defence in pigs. The results seem interesting and intriguing. However, according to the following comments, there are a few methodological concerns that have to be addressed by the authors.

  1. Please rephrase the last paragraph of the Introduction that describes the purpose of the study since it is not clear why this experiment was carried out.
  2. How did the authors choose to include 8 animals in each group? Did they perform a power analysis? If yes, on the basis of which biomarker? If no, this is a serious methodological flaw of the study and should be mentioned.
  3. The experimental design has not been clearly described. For instance, it is not clear to me what are the characteristics of group 3, thermo-neutral (20 °C) and pair-fed TN-pf to HS). Please also add a figure to depict it.
  4. DPPH, ABTS and FRAP are in vitro assays. DPPH and ABTS are commercially available free radicals that do not exist in organisms. So, what is the biological meaning of performing these measurements?
  5. Why did the authors choose the specific muscles?
  6. I suppose that the authors designed an experiment to simulate an environment relevant one anticipated due to climate change. However, it is not explained why the heat stress conditions refer to 30 degrees Celsius. It seems arbitrary. Please explain.
  7. The results are interesting since, probably unexpectedly, heat stress does not appear to be harmful in terms of oxidative stress. I suggest that the authors build a table to include all other relevant studies they cite and compare parameters such as study design, number of subjects, duration of intervention, measured biomarkers etc. Thus, they will possibly have the ability to explain their findings.
  8. Are there any proposed molecular mechanisms regarding the adaptability of the animals to heat stress?

Author Response

Reviewer 2

-In this study, the authors examined the effects of heat stress on oxidative stress and antioxidant defense in pigs. The results seem interesting and intriguing. However, according to the following comments, there are a few methodological concerns that have to be addressed by the authors.

Thank you very much for your comments and suggestions.

-Please rephrase the last paragraph of the Introduction that describes the purpose of the study since it is not clear why this experiment was carried out.

The last paragraph of the Introduction has been rewritten trying to better expose the main objective of the study (lines 64-67).

-How did the authors choose to include 8 animals in each group? Did they perform a power analysis? If yes, on the basis of which biomarker? If no, this is a serious methodological flaw of the study and should be mentioned.

The statistical power was calculated using G*Power 3.1 (Faul, Erdfelder, Lang, & Buchner, 2007, Behavior Research Methods, 39 (2), 175-191). The version 3.1.9.2 from Düsseldorf University was used. On the basis of intramuscular fat (IMF) and lightness (L*) the statistical power for α=0.05 with n=8 was within the range 76-80 %.

-The experimental design has not been clearly described. For instance, it is not clear to me what are the characteristics of group 3, thermo-neutral (20 °C) and pair-fed TN-pf to HS). Please also add a figure to depict it.

Thank you for this question. According to previous bibliography, heat stress usually leads to decrease feed intake in pigs and the possible changes in meat quality could be attributed to the different intake level, rather than to a direct effect of the high temperature. Therefore, to avoid the confounding effects of dissimilar feed intake, a group pair-fed with the heat-stressed group. i.e. under different thermal conditions but at the same feeding level, was included in the study. We have added this justification in the new version of the manuscript.

Following the suggestion of the referee, we have also included a figure of the experimental design (new Figure 1). Numeration or the remainder figures have consequently changed.

-DPPH, ABTS and FRAP are in vitro assays. DPPH and ABTS are commercially available free radicals that do not exist in organisms. So, what is the biological meaning of performing these measurements?

The ABTS, DPPH and FRAP assays are usually used for determining the antioxidant activity of foods, including meat. They measure the capacity of the sample to scavenge free radicals (using in vitro stable radicals) or the reducing power and, therefore, are indicative of the presence of compounds with antioxidant properties (although they do not identity such compounds). It has been shown that enhanced values of such in vitro assays in meat are accompanied by an increase in antioxidant capacity, implicating the decrease of oxidative damage (Hosseindoust et al., 2020, doi:10.3390/antiox9111032).

Previous literature supports the usefulness of the ABTS, DPPH and FRAP assays for evaluating the antioxidant activity in meat, such as those included in our manuscript:

-Echegaray, N.; Munekata, P.E.S.; Centeno, J.A.; Domínguez, R.; Pateiro, M.; Carballo, J.; Lorenzo, J.M. Total phenol content and antioxidant activity of different celta pig carcass locations as affected by the finishing diet (Chestnuts or commercial feed). Antioxidants. 2020, 10, 5.     

-Hosseindoust, A.; Oh, S.M.; Ko, H.S.; Jeon, S. M.; Ha, S.H.; Jang, A.; Son, J.S.; Kim, G.Y.; Kang, H.K.; Kim, J.S. Muscle antioxidant activity and meat quality are altered by supplementation of astaxanthin in broilers exposed to high temperature. Antioxidants. 2020, 9, 1032-

-Tejerina, D.; García-Torres, S.; Cabeza de Vaca, M.; Vázquez, F.M.; Cava, R. Effect of production system on physical-chemical, antioxidant and fatty acids composition of Longissimus dorsi and Serratus ventralis muscles from Iberian pig. Food Chem. 2012, 133, 293-299.

-Why did the authors choose the specific muscles?

Different reasons justify the choice of muscles longissimus lumborum and gluteus medius in our study. Firstly, they are muscles of different body function and metabolism (glycolytic and oxidative) and offer a useful opportunity of study the response to heat exposure according with fibers nature. Secondly, as we stated in the manuscript (lines 68-70), they are representatives of the commercial pieces more valuables and appreciated by consumers. And finally, they are muscles frequently used by the scientific community in the study of quality traits of meat, which favors the comparison of the results between the different researchers.

-I suppose that the authors designed an experiment to simulate an environment relevant one anticipated due to climate change. However, it is not explained why the heat stress conditions refer to 30 degrees Celsius. It seems arbitrary. Please explain.

Thank you for the question.

It is known that heat stress in pigs occurs when the ambient temperature is higher than the animal’s thermoneutral zone, i.e., the range of environmental temperature within which the animals uses no additional energy to maintain their body temperature (Zhang et al., 2020, doi.org/10.1007/s00484-020-01929-6). Although such temperature may vary depending on breeds, production area and growing stage, the upper limit of the thermoneutral or comfort zone for growing pigs has been established at approximately 25 °C (Renaudeau et al., 2007, doi:10.2527/jas.2006-430). Therefore, most studies dealing with the effect of high ambient temperature on pigs have been performed at around 30 °C (Yang et al., 2014, doi.org/10.5713/ajas.2014.14063;  Xin et al., 2015, doi: 10.1016/S2095-3119(15)61061-9;  Xing et al. 2016, doi: 10.1016/S2095-3119(15)61061-9  and others) or even more (32 ° C, Sanz Fernandez et al. 2015, doi: 10.14814/phy2.12315;  35 ° C, Rosado Montilla et al., 2014, doi.org/10.4161/temp.28844; Shi et al. 2016, doi.org/10.1071/AN15003). The duration of exposure to high environmental temperature vary between studies, depending on simulation of acute or chronic heat stress conditions, that suppose application of heat over a long period of time (days to weeks) (Renaudeau et al., 2021, doi.org/10.1016/j.animal.2021.100372). Thus, as the present study was designed to evaluate the effects of long term exposure to heat stress, 30°C and 28 days were selected as temperature and duration of the experimental period. 

-The results are interesting since, probably unexpectedly, heat stress does not appear to be harmful in terms of oxidative stress. I suggest that the authors build a table to include all other relevant studies they cite and compare parameters such as study design, number of subjects, duration of intervention, measured biomarkers etc. Thus, they will possibly have the ability to explain their findings.

We agree that findings of the present assay were somewhat surprising, especially on the basis of previous bibliography.

Including a table compiling the related bibliography is a good idea and we thank very much the suggestion. However, we think that the “Results and discussion” section in its current form collects all the studies related to our research, in a clear and concise way. A bibliography table seems more suitable for a review, and therefore, if the editor agrees, we prefer to keep the text in its current version, also with the aim of not lengthening the manuscript excessively.

-Are there any proposed molecular mechanisms regarding the adaptability of the animals to heat stress?

The adaptive response to hyperthermia has led to different results depending upon the species. In general, it is accepted that one of the main molecular mechanism involved in the response to heat stress is related with the heat shock proteins (HSP), originally identified as proteins whose expression was markedly increased by heat shock. They are implicated in protecting cells subject to denaturation by heat and in regulation of cellular redox states (Parkunan  et al., 2017, doi.org/10.5713/ajas.16.0020; Eugenev et al. 2007, doi: 10.1007/s12038-007-0048-6). We have included some of these comments in the manuscript (lines 426-431).

In addition, we have suggested in the manuscript other possible mechanisms involved in the response to heat stress, such as the increased mRNA expressions of enzymes participating in the FA synthesis (lines 326-330), which is strongly related with meat quality.

The molecular mechanisms involved in the adaptability of Iberian pigs to heat stress deserve to be studied, and we have included such consideration in the conclusion section of the new version of the manuscript.

Thank you again for the comments and suggestions.